# Direct Rating Estimation of Enlarged Perivascular Spaces (EPVS) in Brain MRI Using Deep Neural Network

**Ehwa Yang [1], Venkateswarlu Gonuguntla [1], Won-Jin Moon [2], Yeonsil Moon [3], Hee-Jin Kim [4], Mina Park [5] and Jae-Hun Kim [6],***

1   School of Medicine, Sungkyunkwan University, Seoul 06351, Korea; ehwayang@gmail.com (E.Y.); venkateswarlu.phd@gmail.com (V.G.)
2   Department of Radiology, Konkuk University Medical Center, School of Medicine, Konkuk University, Seoul 05030, Korea; mdmoonwj@kuh.ac.kr
3   Department of Neurology, Konkuk University Medical Center, School of Medicine, Konkuk University, Seoul 05030, Korea; 20060246@kuh.ac.kr
4   Department of Neurology, College of Medicine, Hanyang University, Seoul 04763, Korea; ewhabrain@naver.com
5   Department of Radiology, Gangnam Severance Hospital, College of Medicine, Yonsei University, Seoul 06273, Korea; to.minapark@gmail.com
6   Department of Radiology, Samsung Medical Center, School of Medicine, Sungkyunkwan University, Irwon-dong #50, Gangnam-Gu, Seoul 138-700, Korea
*   Correspondence: jaehun.kim78@gmail.com; Tel.: +82-2-3410-6494

**Abstract:** In this article, we propose a deep-learning-based estimation model for rating enlarged perivascular spaces (EPVS) in the brain's basal ganglia region using T2-weighted magnetic resonance imaging (MRI) images. The proposed method estimates the EPVS rating directly from the T2-weighted MRI without using either the detection or the segmentation of EVPS. The model uses the cropped basal ganglia region on the T2-weighted MRI. We formulated the rating of EPVS as a multi-class classification problem. Model performance was evaluated using 96 subjects' T2-weighted MRI data that were collected from two hospitals. The results show that the proposed method can automatically rate EPVS—demonstrating great potential to be used as a risk indicator of dementia to aid early diagnosis.

**Keywords:** brain; magnetic resonance imaging; enlarged perivascular spaces; deep learning; dementia

## 1. Introduction

Perivascular spaces (PVSs), also known as Virchow-Robin spaces, are the pial-lined and fluid-filled cavities surrounding the walls of arteries, arterioles, veins, and venules in the brain [1]. PVSs have a significant role in the glymphatic system and waste clearance process [2,3] and are key to regulating immunological responses [4]. Several studies on enlarged perivascular spaces (EPVS) in the brain indicate that their presence is an indication of neuronal dysfunctions and that they could serve as a biomarker to diagnose brain diseases such as dementia [5–7], multiple sclerosis [8], stroke [9–13], Parkinson's [14], and small vessel diseases [15–19]. Due to its soft tissue contrast and high-resolution features, magnetic resonance imaging (MRI) is the most often utilized imaging tool in the assessment of EPVS. In MRI scans, PVSs are observed as well-defined oval/rounded/tubular structures with smooth margins. However, quantification of EPVS is challenging due to the difficulty of discriminating them from lacunes (both lacunes and EPVS have a cerebrospinal fluid (CSF)-like intensity on T1-weighted and T2-weighted MRI and are observed as hyperintensities on T2-weighted MRI and hypo-intensities on T1-weighted MRI scans [20,21]).

In general, clinical studies manually count the number of EPVS presented in a region of interest [20,22–24] (basal ganglia, hippocampus, midbrain, and centrum semiovale) and rate the presence of EPVS on a five-point scale [20]. However, there is an intra- and

inter-reader variability of detecting EPVS due to their tiny structures and morphological ambiguity [20]. In brain disease diagnosis, the manual assessment of EPVS is the key step but is challenging and time consuming due to (i) the difficulty in discriminating the EPVS from brain lesions with similar appearances and (ii) the existence of multiple EPVS in single scans.

Several studies have developed automatic and semi-automatic methods for EPVS segmentation. For instance: an automated EPVS segmentation method based on Haar-like features was proposed in [25]. The authors of [26] proposed an interactive segmentation method based on intensity thresholding, while those of [27] used a Frangi filter to enhance EPVS area and performed segmentation of individual EPVS. Recent studies have begun to apply deep learning to EPVS quantification as the popularity of deep learning algorithms have grown due to their superior performance. For example, [28] proposed a multi-channel and multi-scale strategy approach for segmenting EPVS using 7 Tesla MRI scans, while [29] proposed a GP-Unet model for segmenting EPVS using 1.5 Tesla MRI scans. However, most of these methods focus on the segmentation and the detection of the EPVS, and as a result they often lead to inaccurate EPVS rating scores due to their low segmentation performance (existing segmentation methods do not satisfy the performance parameters). Further, the production of the training dataset for deep learning-based segmentation requires the medical experts to manually label EPVS as masks—a time-consuming and laborious process. This is because as many as 20–50 EPVSs can be contained in a single slice.

In contrast, counting the EPVS as a rating score is significantly simpler. The development of methods for directly rating the EPVS could overcome the mentioned drawbacks in existing segmentation methods. In particular, since clinical decisions are made based on the rating score of EPVS, the direct measure of the EPVS rating score is a more convenient way—it bypasses the labor-intensive segmentation practice. Therefore, to deal with the above problems, we propose the direct estimation model of EPVS rating without segmentation or detection.

Currently, only one study reports a successful segmentation-free quantification of EPVS using a 3D regressive convolutional neural network (CNN) [30]. However, this study used 7.0 Tesla MRI scans for its evaluation. There is a big difference in image quality between 3T and 7T, where the EPVS are much more clearly seen in 7T. To our knowledge, our study makes the first attempt to estimate the rating of EPVS without detection or segmentation using a 2D image-based deep learning model trained on 3.0 Tesla MRI scans. In this paper, we present a method for directly rating the EPVS in the basal ganglia using T2-weighted MRI. To do this, we first apply the preprocessing procedure containing Haar-transformation to the MRI to enhance EPVS visibility. Then, a CNN model is used with manual rating as training labels to estimate the EPVS rating. The rating of EVPS was formulated as a multi-class classification problem.

## 2. Materials and Methods

### 2.1. Subjects Information

A total of 96 subjects' MRI data were collected from two hospitals, Konkuk University Medical Center and Hanyang University Seoul Hospital (July 2018 to August 2019). 76 subsets who reported memory complaints (in either subjective or objective form) were categorized as 'patients'. The remaining 20 individuals—who were younger than 45 and had not shown clinical evidence of neurological or psychiatric symptoms—were categorized as 'normal'. The demographic information of all subjects is as described in Table 1. Information such as diagnosis, age, gender, clinical dementia rating (CDR), CDR (SB), and mini-mental status examination (MMSE) of all the subjects are detailed in Table 1. Neuro psychiatric examination scores of MMSE, CDR, and neuropsychiatric battery test were used to diagnose mild cognitive impairment (MCI) and Alzheimer's dementia (AD) based on the criteria suggested in [31]. Based on these criteria, all 96 subjects were classified as: 8—AD, 4—other dementia, 47—MCI, 17—subjective memory impairment (SMI), and

20 control subjects. In this study, the SMI subjects were considered the 'normal' category, and AD and other dementia subjects were considered the 'dementia' category.

**Table 1.** The demographic information of patients (age, clinical dementia rating (CDR), CDR (SB), and MMSE-mini-mental status examination shown as mean ± standard deviation).

| Characteristics | AD | Other Dementia | MCI | SMI | Healthy |
|---|---|---|---|---|---|
| Age | 76.35 ± 1.98 | 70.00 ± 12.72 | 70.21 ± 5.86 | 67.71 ± 9.18 | 31.05 ± 7.37 |
| Male:Female | 2:6 | 1:3 | 16:31 | 6:11 | 13:7 |
| CDR | 0.625 ± 0.35 | 0.5 ± 0.0 | 0.46 ± 0.14 | 0.29 ± 0.25 | - |
| CDR (SB) | 3.5 ± 1.96 | 1.375 ± 1.11 | 1.27 ± 0.87 | 0.44 ± 0.35 | - |
| MMSE | 22 ± 4.21 | 24.5 ± 3.70 | 26.26 ± 2.58 | 28.85 ± 1.69 | - |

*2.2. Data*

All subjects' MRIs were acquired using SIEMENS 3.0T Skyra (20-channel coil). The protocol includes 3D T1-weighted MRI, 3D/2D FLAIR MRI, and 3D SWI. Data of 76 patients consist of T1, T2, and FLAIR images. Data of 20 normal subjects include T2 and FLAIR images. For T1-weighted images, magnetization prepared rapid gradient echo (MPRAGE) sequence was used with repetition time (TR)/echo time (TE) = 2300 ms/2.98 ms; inversion time = 900; matrix size 256 × 256 in-plane resolution = 1 × 1 mm; slice thickness = 1 mm; number of slices = 192; and Grappa factor = 2. For T2-weighted images, an axial turbo spin-echo sequence was used with repetition time (TR)/echo time (TE) = 4450 ms/81 ms; flip angle = 150 degree; matrix size = 384 × 384; in-plane resolution = 0.573 × 0.573 mm; slice thickness = 5 mm; number of slices = 28. For FLAIR images, the 3D sequences were obtained using repetition time (TR)/echo time (TE) = 5000 ms/393 ms; inversion time = 1800 ms; matrix size = 256 × 256; in-plane resolution = 1 × 1 mm; slice thickness = 1 mm; and number of slices = 196. Parameters used to obtain the 2D FLAIR images include: repetition time (TR)/echo time (TE) = 9000 ms/95 ms; inversion time = 1800 ms; matrix size = 320 × 320; in-plane resolution = 0.688 × 0.688 mm; slice thickness = 5 mm; and number of slices = 28.

*2.3. Manual Assessment of EPVS*

This study defines EPVS as small, sharply delineated structures of less than 3 mm following the movement of the perforating vessel [23] and containing CSF intensity (or near-CSF intensity). Visual assessment of EPVS rating was carried out manually by two neuroradiologists (one with 22 years of experience and the other with 8 years of experience) who were blinded to the clinical diagnosis one month prior to the visual assessment session. The two raters conducted their own analyses and reached a consensus on the final decision. All images (T1-weighted, T2-weighted MRI, and FLAIR) were considered for manual visual assessment. To begin, EPVS is evaluated using T2-weighted MRI. In FLAIR images, even if surrounding hyperintensity is present, it is regarded as EPVS if it exists inside white matter [23]. EPVS is evaluated based on the three slices that contain the basal ganglia, which is above anterior commissure area. As suggested in [20,22], EPVS evaluation also includes sub-insular white matter originating from perforating artery of the insular cortex as the basal ganglia.

Both the size and shapes are critical in assessing the EPVS. When the exact count is difficult to obtain, general impression of imaging features can aid the EPVS rating [20,22]. Thus, the general impression of each rater is also recorded to grade EPVS. Finally, we recorded the counts of EPVS on both brains (three slices) of basal ganglia area independently and selected the highest count for rating. In cases of asymmetrical distribution in each anatomical region, the higher count from the two regions is selected for rating the EPVS. The rating (five point scale [20]) was made based on the EPVS counts (Table 2). Along with the rating score of the EPVS, the manual ratings of the dataset for all 576 images (96 subjects) are also presented in Table 2. Agreement between the two raters (evaluated

via intraclass correlation coefficient (ICC) using the healthy controls' T2 images) were 88.0% (95% confidence interval: 0.761–0.942) and 90.1% (95% confidence interval: 0.800–0.952).

**Table 2.** Rating score of EPVS and manual ratings of the dataset (576 images in total) in basal ganglia region (0 (no EPVS found); 1–10 (mild); 11–20 (moderate); 21–40 (frequent); >40 (severe)).

| EPVS Rating | 0 | 1 | 2 | 3 | 4 |
|---|---|---|---|---|---|
| Number of EPVS | 0 | 1–10 | 11–20 | 21–40 | >40 |
| Number of images | 120 | 150 | 198 | 90 | 18 |

### 2.4. Deep Learning Model for EPVS Rating Estimation

The objective of this study is to directly estimate the rating of EPVS. The proposed framework of EPVS rating estimation comprised two steps and is as illustrated in Figure 1. The first step, denoted by the highlighted box in Figure 1, involves preprocessing the MRI to emphasize the EPVS region for better visibility and resizing it to accommodate the input of the deep neural network. In step 2, which is also indicated by the highlighted box in Figure 1, we estimate the EPVS rating using a convolutional neural network (CNN). We describe the preprocessing methods used in this study and proposed deep neural network for direct rating estimation of EPVS in the sections that follow.

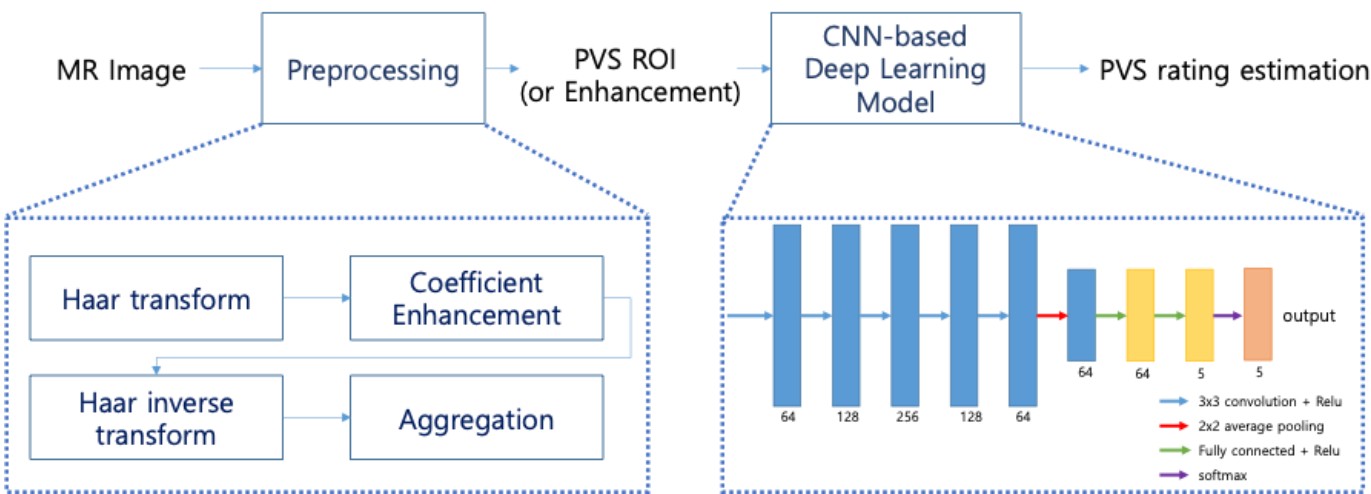

**Figure 1.** Proposed framework for direct rating estimation of EPVS. Each highlighted box represents the image pre-processing to enhance the EPVS region and the EPVS rating estimation model.

#### 2.4.1. Preprocessing

During the preprocessing step, first the signal intensities of MRI were normalized in the range 0–1 by dividing the maximum value for each MRI scan. Then, in three selected MRI slices of the left and right brain, basal ganglia regions were cropped. Cropping the basal ganglia regions was necessary because EPVS ratings were estimated solely based on the basal ganglia region. Overall, a total of 576 images (6 images × 96 subject) were obtained and used as a dataset. Finally, the cropped images were enhanced using a Haar-transform-based approach [32]. The application of Haar-transform to the cropped images resulted in emphasized images with clear visibility of EPVS. For illustration purposes, the cropped image before and after applying Haar-transform to the selected subject is shown in Figure 2. In Figure 2 (circled with red color), we can clearly observe that the EPVS of the basal ganglia region has better visibility. Finally, the pre-processed images were resized to 112 × 86 before feeding to the deep neural network.

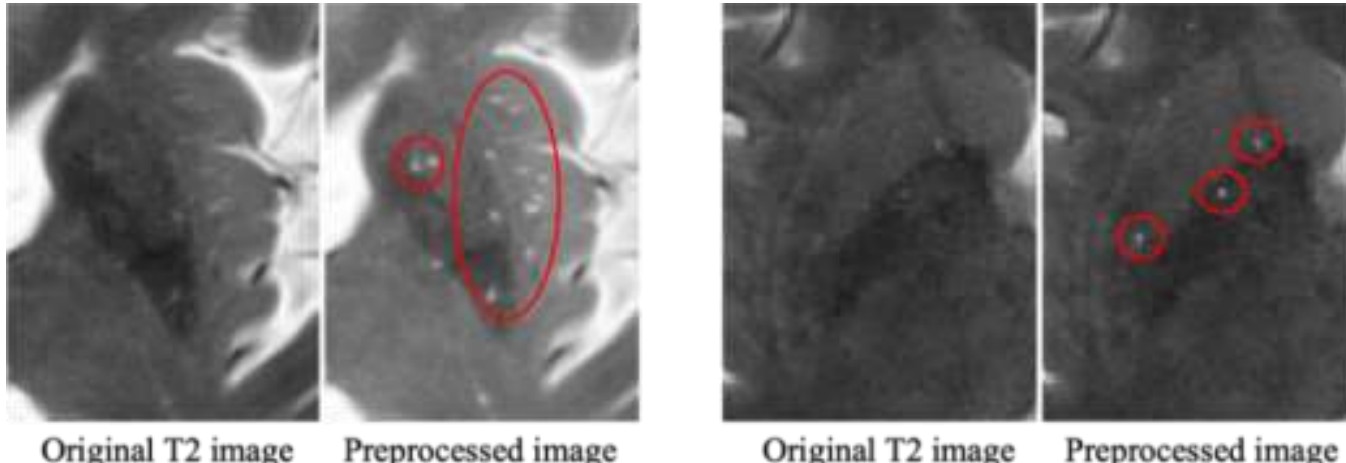

**Figure 2.** The comparison of EPVS in originally cropped image (**left**) and the image after applying the Haar-transformation (**right**) on T2-weighted MRI scans in the basal ganglia region of selected subjects (We can observe that the EPVS (circled in red) are enhanced and have better visibility after preprocessing).

### 2.4.2. Deep Neural Network

As illustrated in Figure 1, the proposed deep neural network for directly estimating the EPVS rating consists of five convolutional layers, an average pooling layer, two fully connected layers, and a classification softmax layer. The final layer, for classification, is composed of five nodes, representing the EPVS' five-point rating scale. The pre-processed image size is provided as an input for the network. Since EPVS have thin and tiny structures, average pooling over the max pooling layer is selected after several convolutions to maximally preserve the EPVS features.

**Training:** The proposed network was trained in an image-by-image manner. We divided the 576 images from 96 subjects into 462 training images from 77 subjects and 114 test images from 19 subjects for training and testing purposes. We used cross-entropy loss function for loss function. Additionally, for another loss function, we trained our model with mean square error (MSE). Adam optimizer with the learning rate of 0.001 was used. The network was trained for 300 epochs. To increase the diversity of the data and for better generalization of the trained network, data augmentation was performed using 2D rotation and flipping (each axis was rotated $-30$ and 30 degrees). The proposed method was implemented on TensorFlow 1.14 using a PC equipped with four GPUs (NVIDIA Geforce RTX 2080Ti).

**Testing:** During the testing phase, six images per subject were fed into the model and we obtained six EPVS ratings (EPVS rating score at image level). Finally, the highest score among the six images was selected for the EPVS rating estimation.

### 2.5. Evaluation for Deep Learning Model

We assessed our model's accuracy at the image and subject levels. In terms of methodology, since we trained our deep learning model in a slice-by-slice manner, we reported the accuracy of our model at the image level. However, because one subject was evaluated using several MRI slices in a clinical setting, we used majority voting to report the accuracy of our model at the subject-level.

The training and testing data were carefully selected to evenly include all groups, i.e., dementia (AD and other dementia), MCI, and normal (SMI, healthy) subjects. Using five-fold cross validation, the average estimation score is reported as the model accuracy. Furthermore, the performance of the model for individual groups (dementia, MCI, and normal) was also analyzed and reported. Finally, the correlation between the model estimation and ground truth results (both image level and subject level) were analyzed using Spearman's correlation coefficient.

## 3. Results

Using the data split as described previously, the proposed model was trained with 77 subjects and tested with 19 subjects. Table 3 summarizes the model's performance at the image and subject levels. A training accuracy of 88.1% at image level and 83.6% at subject level was observed. Overall, an accuracy of 87.7% at image level and an accuracy of 80.2% at subject level were achieved. The performance of the model for individual groups are also tabulated in Table 3. In the case of dementia, the model achieved an accuracy of 91.7% at both image and subject level. In case of MCI, we obtained an accuracy of 83.0% at image level and 80.9% at subject level. For healthy subjects, we observed an accuracy of 92.3% at image level and 75.7% at subject level.

**Table 3.** Performance of EPVS rating estimation model on whole data and for individual groups (testing and training accuracy is reported as mean $\pm$ standard deviation).

| Accuracy (%) | | Whole Data | | For Individual Groups | | |
| --- | --- | --- | --- | --- | --- | --- |
| | Loss | Training | Testing | Dementia | MCI | Healthy |
| Image level | Cross-entropy | 88.1 $\pm$ 0.6 | 87.7 $\pm$ 1.6 | 91.7 | 83.0 | 92.3 |
| | MSE | 87.1 $\pm$ 0.9 | 87.6 $\pm$ 2.7 | 88.7 | 83.1 | 90.3 |
| Subject level | Cross-entropy | 83.6 $\pm$ 1.5 | 80.3 $\pm$ 6.6 | 91.7 | 80.9 | 75.7 |
| | MSE | 82.9 $\pm$ 2.9 | 80.1 $\pm$ 4.5 | 87.9 | 78.9 | 76.5 |

Spearman's correlation coefficient between the estimated rating and ground truths at both image level and subject level were evaluated and are as shown in Figure 3. A correlation coefficient of 0.9169 ($p$-value < 0.0001) at subject level and 0.9335 ($p$-value < 0.0001) at image level shows the reliability of the estimated values. Subject wise prediction of all subjects considered in this study are also analyzed and the results are as shown in Figure 4. In Figure 4, the estimated rating score of each subject, along with the ground truth, is presented. Figure 4a depicts all subjects who were correctly predicted and Figure 4b depicts all subjects who were incorrectly predicted. In the case of correctly predicted subjects, the prediction was correct despite the differences presented in the image-wise predictions. In the case of incorrectly predicted subjects, the prediction scores of most of the subjects were lower than the manual score (ground truth). This is because the numbers of EPVS were close to the critical numbers of the decision boundaries (for instance, when the number of EPVS is 11 or 21, the model tends to estimate the EPVS rating score as 1 or 2 instead of 2 or 3).

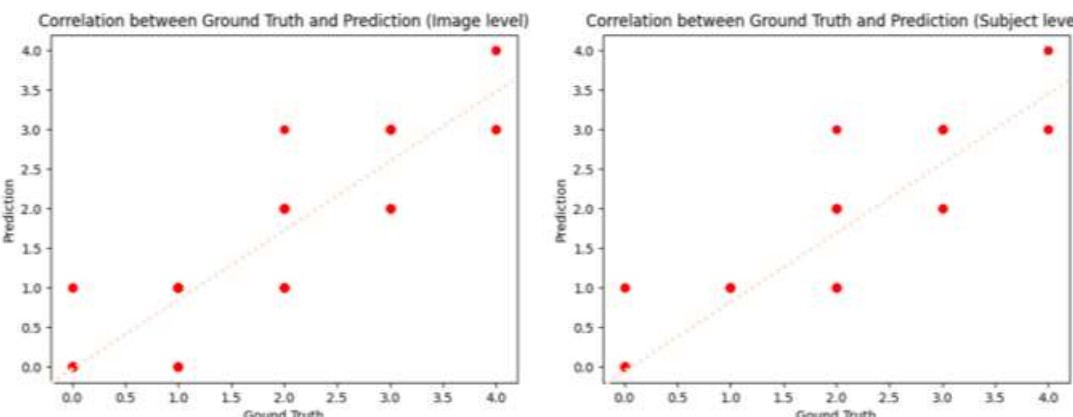

**Figure 3.** Spearman's correlation between the predictions and the ground truth rating scales at image level (**left**) and at subject level (**right**) (at image level, all points (Gi, Pi) $\forall$ i = 1:576 and at subject level, all points (Gs, Ps) $\forall$ s = 1:96 were shown in the figure where G is the ground truth and P is the prediction. Spearman's correlation coefficient of 0.9169 ($p$-value < 0.0001) at subject level and 0.9335 ($p$-value < 0.0001) at image level shows the reliability of the model estimation.

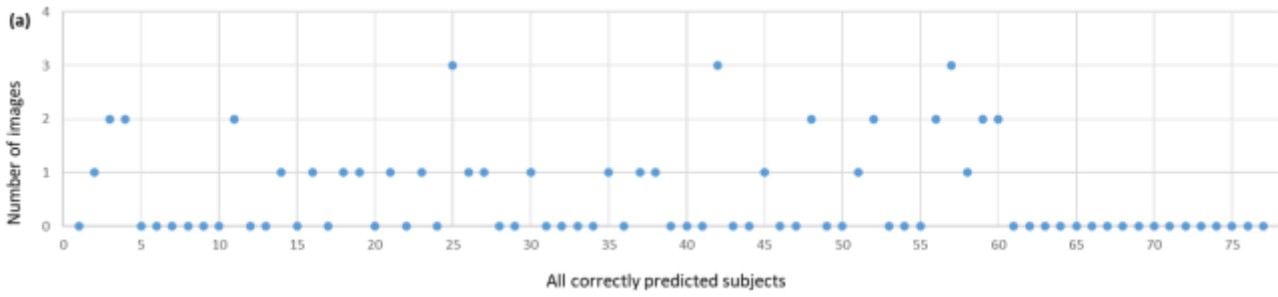

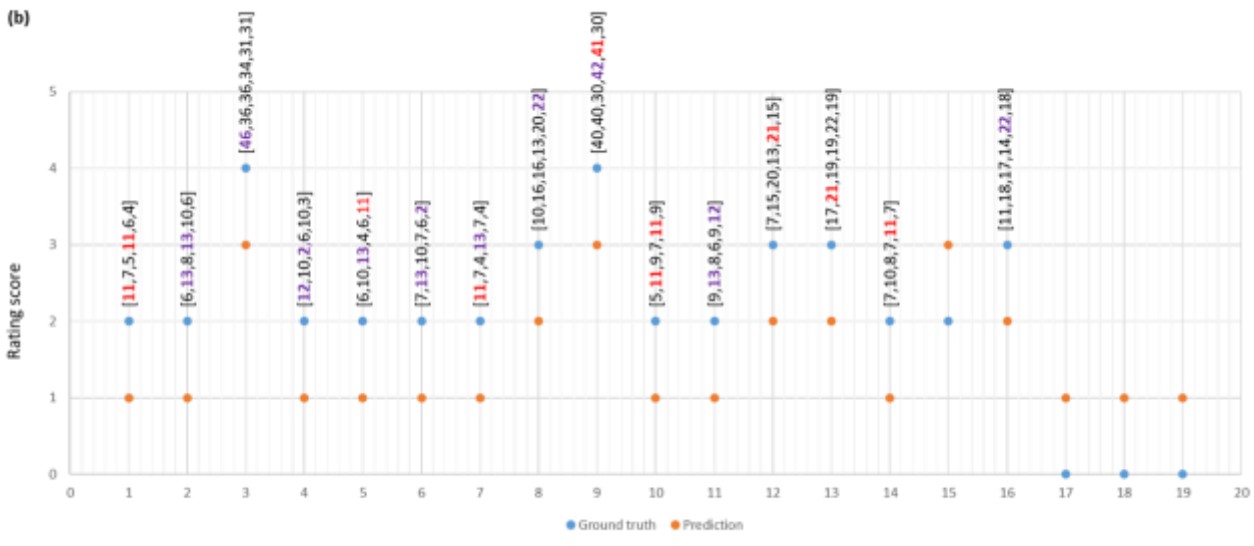

**Figure 4.** The prediction results of individual subjects: (**a**) All correctly predicted subjects (vertical axis; the number of images indicates the number of incorrectly predicted images per subject (out of six images per subject)). At subject level, all these subjects were rated correctly, although there were some incorrect predictions at image level. (**b**) All incorrectly predicted subjects (numbers corresponding to each subject in the brackets shows the number of EPVS found in each image). The numbers highlighted red and purple are the incorrect EPVS numbers identified in the corresponding image that have affected the final rating of the subject. We can also observe that the numbers highlighted in red are close to the decision boundary and caused under rating of EPVS when compared to ground truth.

Table 4 summarizes the findings of an ablation study (an experiment with and without the pre-processing step). In pre-processing, the Haar transformation-based approach was applied to enhance the EPVS in MRI images. When the pre-processing step is included, estimation performances are increased by 10% at the subject level and 11% at the image level. This experiment result demonstrates that pre-processing for EPVS enhancement results in a more robust estimation model.

**Table 4.** Results of an ablation study on the performance of the EPVS rating estimation model with and without pre-processing with the Haar transformation approach (testing and training accuracy is reported as mean ± standard deviation; cross-entropy was used for loss function during training).

| Accuracy (%) | | Whole Data | | For Individual Groups | | |
|---|---|---|---|---|---|---|
| | | Training | Testing | Dementia | MCI | Healthy |
| Image level | No pre-processing | 78.3 ± 0.9 | 76.2 ± 1.7 | 80.2 | 72.3 | 80.6 |
| | Proposed procedure | 88.1 ± 0.6 | 87.7 ± 1.6 | 91.7 | 83.0 | 92.3 |
| Subject level | No pre-processing | 72.9 ± 1.9 | 70.2 ± 7.2 | 79.1 | 63.9 | 72.5 |
| | Proposed procedure | 83.6 ± 1.5 | 80.3 ± 6.6 | 91.7 | 80.9 | 75.7 |

We compared our model to the automatic method of EPVS quantification using Haar-like features and random forest as a baseline approach (see [25]). The compared method is a traditional approach that is one of the most represented techniques for the quantification of EPVS. We trained and tested the baseline method with the same dataset used in our estimation model. Compared to the other study, our estimation model has ~17% increased accuracy at both image and subject level (the baseline approach showed 69.73 ± 0.5% accuracy at image level and 63.02 ± 1.4% accuracy at subject level for our dataset).

## 4. Discussion

In this study, we proposed a framework to directly estimate the EPVS rating from T2-weighted MRI without using either detection or segmentation of EVPS. The overall accuracy of 87.7% at image level and 80.2% at subject level with the proposed method shows that the EPVS rating can be directly estimated from T2-weighted MRI images. We implemented our method following the same procedure as radiologists do. In clinical practice, radiologists choose the EVPS score based on the highest number of EPVS across multiple slices of MRI. As a result, we implemented this decision process as a majority vote using the results of image-level predictions. Since we used majority voting in the subject level prediction, the performance of the model at subject level was less accurate when compared to its performance at image level. We observed this decrease in performance because the correct prediction at the image-level in the presence of a low EPVS score could not be considered at the subject-level prediction.

Individual analyses of each subject's EPVS rating prediction reveals that for incorrectly predicted subjects the prediction score was lower than the manual score in most subjects (15 out of 19). One possible explanation is that the number of subjects presented in the range of [0, 2] is higher than the number of subjects presented in the range of [3, 4]. Another reason may be that many of the EPVS numbers were close to the critical number of the decision boundary and may have caused under-rating.

We can see from the result analysis of each group that the predictions of all groups at image level are like the predictions at subject level, except for normal subjects (92.3% at image-level, and 75.7% at subject-level). This discrepancy was caused by the strict criteria of rating at zero level (zero means that there are no EPVS in any of the images corresponding to a particular subject. Additionally, we used majority voting from the six images and predicted the score at the subject level). More sophisticated methods of voting could nullify this gap between subject level and image level.

This study has a limitation for comparison with other state-of-the-art methods due to its different labeling scheme. We labeled the EPVS as the score (weak labeling), but for comparison with segmentation networks we must label the EPVS as masks of MRI slices (strong labeling), in which there are 20–50 EVPS (About 50% or more of the total data) in one slice. Moreover, one existing method for EPVS quantification is 3D modeling, but in our case we used a 2D model. We followed Potter et al.'s method [20] for making the labeling as widely applicable to clinical practice as possible and this means using 2D slice-based measurements of portions of the basal ganglia region in MRI images. The existing 3D models use the whole basal ganglia region of MRI images. Therefore, we simply compared our model with the traditional method as a baseline approach and of the two our model gave significantly better performance.

While doing manual ratings of EPVS, we first identified EPVS on T2-weighted MRI images and then confirmed this with T1-weighted MRI and FLAIR scans. However, the study includes only the T2-weighted MRI in the analysis. This is due to the variations in the acquisition parameters such as direction of acquisition, image matrix, FOV, and slice thickness. Though the T1-weighted MRI and FLAIR could provide additional features and improve the accuracy further, it is not convenient to apply the image registration between T1-weighted, T2-weighted, and FLAIR images. Thus, although T2-weighted MRI alone is used to directly predict the rating of EVPS, we obtained comparable accuracy to state-of-the-art methods at both image- and subject-level.

Direct grading of the EPVS rather than detection or segmentation is more convenient and efficient in clinical practice. Since the model estimates the rating directly from T2-weighted MRI without either detection or segmentation of EPVS, the proposed framework has great potential for use as a disease risk indicator. However, before the proposed framework can be used in clinical practice, the EPVS rating estimation model must be further improved with sufficient data training. In the future, we intend to continue collecting data in order to develop an efficient model for estimating the EPVS rating accurately.

## 5. Conclusions

In this paper we proposed a deep learning model for directly estimating the EPVS rating from T2-weighted MRI images without requiring detection or segmentation. The proposed model is trained in an image-by-image manner and validated at the subject-level via the majority voting method. Since the proposed model followed the same procedure as a radiologist does and estimates the EPVS rating directly from T2-weighted MRI images without either detection or segmentation, it is convenient and efficient for clinical use. Our results demonstrate that the proposed method can be used as a disease risk indicator, assisting in the early detection of dementia. This paper may be of interest to readers who are interested in neurology and its applications.

**Author Contributions:** Conceptualization, J.-H.K. and E.Y.; methodology, E.Y.; software, E.Y.; validation, E.Y., M.P.; formal analysis, E.Y., J.-H.K. and W.-J.M.; investigation, E.Y., V.G. and W.-J.M.; resources, E.Y., V.G. and J.-H.K.; data curation, W.-J.M., Y.M. and H.-J.K.; writing—original draft preparation, E.Y.; writing—review and editing, V.G. and J.-H.K.; visualization, E.Y. and V.G.; supervision, J.-H.K.; project administration, J.-H.K.; funding acquisition, J.-H.K. and W.-J.M. All authors have read and agreed to the published version of the manuscript.

**Funding:** This research was supported by a grant from the Korea Health Technology R&D Project through the Korea Health Industry Development Institute (KHIDI), which is funded by the Ministry of Health & Welfare, Republic of Korea (grant number: HI18C1038 and HU21C0222).

**Institutional Review Board Statement:** The study was approved by the Institutional Review Board (or Ethics Committee) of Konkuk University Medical Center (CRIS registration number KCT000318, date of approval: 2018.07.15.).

**Informed Consent Statement:** Informed consent was obtained from all subjects involved in the study.

**Conflicts of Interest:** The authors declare no conflict of interest.

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
