# Peer review of "Direct Rating Estimation of Enlarged Perivascular Spaces (EPVS) in Brain MRI Using Deep Neural Network"

_applsci, doi:10.3390/app11209398_

Round 1
Reviewer 1 Report
- The present state of the article requires extensive format and style correction, and language editing to improve the presentation and user’s readability.
- Avoid using abbreviated words before they are expanded appropriately. For instance, MRI.
- The article seems to be misaligned few statements, figures, and tables. Using figures and tables appropriately with their respective content section shall be considered.
- The introduction section looks very much incapacitated. More insights shall be added with a related study. Extending the introduction section with clearly mentioning the problem statement and the proposed solution in this work shall be considered.
- The contributions made in this work are unclear. Statements to clearly highlight the novelty of the work shall be considered.
- What is the protocol considered for data curation in section 2.2?
- The result section seems to be inadequate. The prediction accuracy achieved at image level and at the subject level is not substantiated with conclusive evidence and also this will demand a comparative study with similar work.
- Overall, the readability of the article shall be further improved with formatting corrections and more insights shall be provided with a related study along with an introduction section and a comparative study in the results section.
Reviewer 2 Report
This work addresses the rating of Enlarged Perivascular Spaces (EPVS) in brain MRI by means of a 2-steps processing pipeline: an image pre-processing based on Haar transform (similar to [31]) and a Convolutional Neural Network (CNN), which is trained to solve a multi-classification problem (5 classes, i.e. EPVS rates from 0 to 4). The novelty is limited but the application is interesting since the method needs no prior segmentation. Performance is assessed on a proprietary dataset. No performance comparison is carried out.
The lack of performance comparison with other approaches is my major concern about this work.
Since a proprietary dataset is used, in order to correctly assess the performance, at least one baseline approach and one state-of-the-art approach should be implemented and tested using the very same training and testing data. The state-of-the-art approach can include a segmentation step, if necessary.
In addition, the authors should perform an ablation study. As an example, a performance comparison of the proposed approach both with and without the pre-processing step would be useful.
Other comments:
- From Tab. 1 it is clear that there is a huge gap between the age of healthy subjects and that of other patients. Does this bias affect performance evaluation?
- The authors state that "Since we used majority voting in the subject level prediction, the performance of the model at subject level was less when compared to performance of the model at image level". Please, better explain why this happens and, if the reason for this performance drop is the majority voting, then why did you use it? Using more data should yield better performance.
- Better explain how "intensities of MRI are normalized in the range [0, 1]". Is this done with respect to the image maximum value? Or computing the maximum intensity value of the whole dataset?
- In the CNN description, the authors state that just one average pooling is adopted. However, in Fig.1 it seems that each convolutional layer is followed by an average pooling. Please clarify and improve the quality of Fig.1 to make the legend more readable.
Suggestions:
- To improve the novelty contribution of this work, consider adopting different loss functions when training the CNN.
Round 2
Reviewer 1 Report
The changes made are noticeable. Extensive language editing shall be considered to improve the readability.
Reviewer 2 Report
The authors addressed most of the comments of this reviewer.
My only remaining concern regards the answer R2-Q1). The authors did not implement any comparison with other techniques after the first review round. Since they use a proprietary dataset, this is very important to assess the performance of the proposed approach.
I understand that other state-of-the-art approaches use different labeling information, however, a comparison can be carried out taking into account this.
Otherwise, the authors should compare performance with at least one baseline approach using the very same training information as the proposed approach.
